# Microbial Signatures Mapping of High and Normal Blood Glucose Participants in the Generation 100 Study

**DOI:** 10.3390/microorganisms13112582

**Published:** 2025-11-12

**Authors:** Natalia G. Bednarska, Line Skarsem Reitlo, Vidar Beisvag, Dorthe Stensvold, Asta Kristine Haberg

**Affiliations:** 1London School of Hygiene and Tropical Medicine, Faculty of Infectious Tropical Diseases, Keppel Street, London WC1E 7HT, UK; 2Department of Neuromedicine and Movement Science, Faculty of Medicine and Health Sciences, Norwegian University of Science and Technology, 7030 Trondheim, Norwayasta.haberg@ntnu.no (A.K.H.); 3Genomics Core Facility, Department of Clinical and Molecular Medicine, Norwegian University of Science and Technology (NTNU), 7491 Trondheim, Norway; 4Central Staff, St. Olav’s University Hospital, 7006 Trondheim, Norway; 5Cardiac Exercise Research Group, Department of Circulation and Medical Imaging, Faculty of Medicine and Health Sciences, Norwegian University of Science and Technology, 7489 Trondheim, Norway

**Keywords:** gut microbiome, type 2 diabetes mellitus, *Akkermansia muciniphila*, Generation 100 Study, probiotics, personalized microbiome

## Abstract

Intestinal dysbiosis has been linked to metabolic disorders, including insulin resistance and type 2 diabetes mellitus (T2DM). T2DM typically follows a prediabetic stage, during which insulin resistance develops. During the early stages of T2DM, its development can be corrected, thus potentially preventing or delaying the onset of the disease. This secondary, exploratory, cross-sectional comparison study aimed to contrast the gut microbiome of individuals with elevated fasting blood glucose to that of individuals with glucose levels within the normal range. This study involved 65 older adults (ages 76–83 years) enrolled from the randomized controlled trial entitled the “Generation 100 Study”, all of whom consented to provide their gut microbiome samples. We employed a high-throughput sequencing of the bacterial 16S rRNA gene to obtain metagenomic microbial profiles for all participants. These profiles were then correlated with clinical measures. Overall, microbial alpha diversity was significantly reduced in the high glucose group. We have also observed distinct patterns of microbial beta diversity between high and normal glucose groups. At the phylum level, we found that Synergistes, Elusimicobia, Euryarchaeota, Verrucomicrobia, and Proteobacteria were all significantly decreased in participants with high blood glucose. Additionally, *P. copri* (ASV 909561) was significantly elevated (10-fold increase) in the high glucose groups, suggesting that it may serve as an early T2DM marker. In contrast to prior reports on the *Fusobacterium* genus, we found that it was significantly increased in the normal glucose group, with a significant 151-fold increase compared to the high glucose group. Directly linking gut microbiota profiles with clinical indicators such as fasting blood glucose and T2DM diagnosis allows the identification of specific microbial features associated with glucose dysregulation, providing preliminary population-level evidence to guide future translational research. Our results indicate significant changes in the microbiome that may provide valuable insights for early intervention in pre-diabetic states.

## 1. Introduction

Type 2 diabetes (T2DM) is a growing public health concern with serious complications, including kidney damage, blindness, cardiovascular disease, and mortality. It is an age-related metabolic disease often considered inflammatory, as various cytokine profiles are associated with its progression [1,2]. A strain of *Prevotella copri*, which has the capacity to produce branched-chain amino acids, was more commonly observed in the gut microbiome of people affected with type 2 diabetes [3]. The circulatory levels of these branched-chain amino acids were linked with a higher risk of T2D [4]. Unequivocally, the decade of research on the gut microbiome has clearly shown common patterns in microbial species depleted or increased in prediabetic and Type II diabetic people compared with healthy individuals [5]. Interestingly, it is difficult to establish cause and effect when it comes to the gut microbiome and insulin resistance development. One possibility is that dietary choices encourage the growth of less beneficial bacterial strains that rely on dietary by-products to thrive. Alternatively, a person may become colonized with unfriendly bacteria that shift the microbiome balance toward producing unhealthy metabolites, which in turn can influence dietary preferences through gut–brain signaling. Over the past decade, extensive research has revealed consistent gut microbial signatures associated with prediabetes and T2DM, including depletion of beneficial taxa such as *Akkermansia* and *Faecalibacterium*, and enrichment of potential pro-inflammatory taxa such as *Prevotella* and *Fusobacterium* [6]. However, establishing causality between the gut microbiome and insulin resistance remains challenging [7]. It is unclear whether dietary factors first alter microbial composition, promoting metabolically unfavorable taxa, or whether pre-existing microbial imbalances influence dietary behavior and host metabolism via gut–brain signaling.

Despite the advances, the relationship between gut microbiota and glycemic control in older adults remains poorly understood. Aging is accompanied by physiological changes that affect both metabolic flexibility and gut microbial diversity, including reduced short-chain fatty acid production, medication use, and dietary shifts [8,9]. Most previous microbiome–diabetes studies have focused on middle-aged adults or individuals with established T2DM, whereas early microbial correlates of mild hyperglycemia in older populations have received limited attention. Understanding these associations could help identify early microbial markers of metabolic decline and opportunities for preventive intervention. Therefore, this study was designed as an exploratory, hypothesis-generating analysis to identify preliminary gut microbial patterns associated with hyperglycemia in an older adult population.

Here we present a secondary analysis from an exercise intervention study in which we invited men and women born between 1936–1942 (n = 73) to a randomized 5-year, twice weekly, either high intensity interval training (HIITI or moderate/medium intensity continuous training (MICT) sessions, or a control group following physical activity as per national recommendations [10]. Participants in the HIIT group were asked to perform two weekly exercise sessions of 40 min duration each. Participants in the moderate intensity continuous training group (MICT) were asked to perform 2 weekly exercise sessions of 50 min [11]. Participants were subjected to clinical examinations, physical tests, questionnaires, and gut microbiome sampling after 5 years at the end of the exercise intervention. Given the known connections between the gut microbiome and insulin sensitivity, we sought to determine whether increased fasting glucose levels had any impact on the microbiome. We hypothesized that distinct microbial signatures and diversity patterns would be associated with higher fasting glucose levels, reflecting early dysglycemic changes in this aging population. The primary objective was to characterize gut microbiota composition and diversity in relation to glycemic status, thereby providing insight into potential microbial markers of hyperglycemia in older adults. The study design graphical representation can be seen on Figure 1. 

## 2. Materials and Methods

### 2.1. Population, Randomization, Ethics

Older adults participating in the neuroimaging part of the Generation 100 Study were asked if they were interested in taking part in a gut microbiome study at the time of brain magnetic resonance imaging (MRI) after 5 years of exercise intervention. The participants in the G100 Study were invited by mail in letters sent to 6966 adults (3721 women) born between 1936–1942 and registered in the Norwegian National Population Registry with a permanent home address in Trondheim municipality. Of these, 1790 showed an interest. The exclusion criteria were the presence of somatic or psychiatric disease precluding exercise intervention or inclusion in other exercise training studies. Dementia at baseline or diagnosed during the study was an exclusion criterion [10]. A total of 1790 people were interested, of these 223 withdrew, resulting in 1567 participants included. Before randomization, the participants were informed of the possibility of participating in a neuroimaging study during the RCT. Exclusion criteria for the MRI study were limited to standard MRI contraindications (e.g., implanted electronic medical devices) and brain pathology, which could interfere with image analysis. In total, 105 participants were included in the MRI study. After 5 years, 85 participants remained in the study and were asked if interested in taking part in a gut microbiome study. A total of 73 participants agreed to provide a sample of stool, and 65 participants also supplied their blood for analysis (Figure 2).

Randomization of the participants in G100 was performed by the Unit for Applied Clinical Research. The participants were randomized 2:1:1, stratified by sex and cohabitation status (living with someone versus alone) into exercise according to the national guidelines (i.e., >30 min daily moderate physical activity) (control group, n = 780) [12], or supervised exercise with either MICT (n = 387) or HIIT (n = 400) [11]. This study was thus a secondary analysis in the neuroimaging sub-study of the RCT. In this study, we have grouped the participants’ microbiome data according to their glucose status (above or below the normal range, cut off 5.4 mmol/L). To evaluate potential confounding by exercise, we compared the distribution of exercise groups (high-intensity interval training [HIIT], moderate-intensity continuous training [MICT], and control) between glycemia categories using a chi-square test. Exercise group distributions did not differ significantly between normoglycemic and higher-risk participants (χ^2^(2) = 1.86, *p* = 0.39) (see Appendix A). Given this balance, formal multivariate adjustment for exercise status in the differential abundance analysis was not deemed necessary.

The sub-study was approved by the Regional Committee for Medical Research Ethics, Central Norway (2012/849). All participants were legally competent and gave their written informed consent. Descriptive statistics for the key metabolic blood measures, along with glucose level group distribution, can be found in Appendix A.

Participants included in the microbiome study were then divided based on measured fasting blood glucose levels. Participants with glucose levels between 4–5.4 nmol/L were assigned to a normal group, whilst participants with glucose levels higher than >5.5 were assigned to a high glucose group. Based on Cohen’s d value, it was estimated that there’s roughly an 89% chance a random person from the higher-mean group has a higher value than a random person from the lower group (common-language effect size). There were six individuals clinically diagnosed with type 2 diabetes in the high glucose group.

### 2.2. Gut Microbiome Collection and Preparation

The gut microbiome was collected using the same methods as in the National Institute of Health human microbiome project. A polystyrene box (28 × 28 × 28 cm) with a lid was filled with an Fecotainer (UN3373 Medical Packaging), 10 pre-frozen flat, standard cooler elements in a plastic bag, and a zip-lock plastic bag for the used Fecotainer. The lid of the box was secured with a strap. An illustrated instruction on the procedure for providing the stool and packing the box was provided. In addition, the abdominal health and diet questionnaire and a contact phone number for pick-up were included. In the afternoon of day 1, the prepackaged box was delivered by car directly to the participant’s residence. The participant was asked to remove the cooler elements from the box into a freezer awaiting stool sampling on day 2. The participant performed the stool sampling of the entire stool specimen at his/her usual time of defecation. The box was subsequently packed with cooler elements and the Fecotainer according to the instructions, and immediately picked up by a driver who delivered the box to the hospital. Upon arrival at the hospital, the specimens were promptly transferred to a −80 °C freezer. When samples from all participants were acquired, further stool processing commenced. Stool samples were collected from participants between January and March 2019. Immediately after defecation, each sample was placed on ice and transported to the laboratory within 2–3 h. Upon arrival, samples were aliquoted and stored at –80 °C until further processing. All DNA extractions were performed within one week after the final sample collection, and samples remained frozen for no longer than three months before extraction.

### 2.3. DNA Extraction and 16s Ribosomal RNA Sequencing

Total DNA extracts of 73 human stools were performed according to the Standard Operating Procedure published by IHMS Consortium [13,14]. Total cellular DNA was extracted from 0.2 g of starting fecal material using the DNeasy PowerSoil Kit, as per the modified manufacturer protocol (QIAGEN). To improve cellular lysis, the samples were transferred to tubes pre-filled with 750 μL of PowerBead Solution (0.5 mm diameter ceramic/garnet beads) and homogenized using a cell homogenizer (Precellys24, Bertin). The DNA was extracted successfully in sufficient quantity and quality for sequencing 73 samples. (DNA concentration was quantified by using the Qbit method.) 16S metagenomic sequencing libraries were prepared from DNA samples according to the “16S Illumina Demonstrated Library Prep Guide with minor adjustments [15]. In brief, 12.5 ng genomic DNA from the stool samples was used as a template for PCR amplification (25 cycles) of the 16S V3 and V4 regions. The 16S ribosomal RNA gene PCR primers were based on sequences first published by Klindworth et al. [16]. Illumina adaptor compatible overhang nucleotide sequences were added to the gene/locus specific sequences (16S Amplicon PCR Forward Primer=5′TCGTCGGCAGCGTCAGATGTGTATAAGAGACAGCCTACGGGNGCWGCAG and 16S Amplicon PCR Reverse Primer=5′GTCTCGTGGGCTCGGAGATGTGTATAAGAGACAGGACTACHVGGGTATCTAATCC), resulting in a PCR product of the expected size of about 550 bp. The PCR products were then cleaned up by using AMPure XP beads to purify 16S V3 and V4 amplicons away from free primers and primer dimer species. In a second PCR amplification step (8 cycles), dual indices and Illumina sequencing adaptors were added by using the Nextera XT indexing kit (Illumina Inc.) according to the manufacturer’s instructions. A second PCR clean-up step was performed using AMPure XP beads, before validation of the library by a LabChip GX DNA high-sensitivity assay (PerkinElmer, Inc.). Libraries were normalized and pooled to 10 pM and subjected to clustering on one MiSeq V3 flowcell. Finally, paired-end read sequencing was performed for 2 × 300 cycles on a MiSeq instrument (Illumina, Inc., San Diego, CA, USA), according to the manufacturer’s instructions.

### 2.4. Data Quality Control and Analysis

The data were processed with MiSeq Illumina instrument control software (version 3.1) for two-step PCR pools. This step includes MID sequence tag separation and computation of quality control scores and GC distribution. Barcoding was used to demultiplex the samples. FastQC analysis with Illumina built-in software indicated high-quality reads (both reverse and forward) and resulted in good assembly of paired reads later in CLC Genomics software (CLC Genomics Workbench 20.0.3)

For taxonomic classification (clustering), reads were analyzed using the CLC Genomics workflow. Reads were grouped into operational taxonomic units (OTUs) using Greenegenes v13_8 99% version, and the OTUs were matched with a similarity of 99% of the database. The parameters for minimum occurrences of combined reads were set to 50 to reduce the number of false positive matches. New OTUs were recorded to explore the number of unique unmapped sequences (e.g., Chao index for unique OTUs). Forward and reverse reads were matched through the assembly process, and a merge pair report was created with 19,432,643 reads. Chimeras were filtered out of the analysis, at the crossover cost set to the value of 3. After sequence assembly, quality control, rarefaction, and frequency-based filtering, we proceeded to microbiome analysis with 19,432,643 total reads, which allowed us to predict 2646 total OTUs, 395 de novo OTUs, and 585 unique chimeric reads.

The proportion of the gut microbiome at phylum, genus, and species taxonomic levels was determined using the Greene genes database (gg_13_8_otus/taxonomy/97_otu_taxonomy) and described as ratios, relative abundance, and raw numeric abundance (numbers of OTUs). Rarefaction has been done to adjust for differences in library sizes across samples and allow comparisons for ratios and alpha diversity [17,18]. To test for equality of variance among groups, we performed a permutational analysis of multivariate dispersions (PERMDISP) using 999 permutations. Distances from each sample to its group centroid were calculated based on Euclidean distances for Shannon and Simpson diversity indices. F-statistics and permutation *p*-values were obtained to assess whether within-group dispersions differed significantly among glucose or exercise groups (see Appendix A). No significant dispersion effects were detected, supporting the validity of subsequent nonparametric comparisons.

A phylogenetic tree was obtained from the top 100 most abundant OTUs, which were aligned using the MUSCLE algorithm built in CLC Genomics software to reconstruct the phylogeny tree. The maximum likelihood phylogenetic tree was used to measure alpha diversity and beta diversity. The (α)Alpha diversity, a measure of species richness and evenness within each individual sample, was estimated with the Shannon entropy and Simpson diversity indices at the phylum and genus level [19]. The Shannon entropy index summarizes the range of a population in which each member belongs to a unique taxonomic group. The index corresponds directly to sample heterogeneity. The Simpson diversity index measures community diversity considering the number of taxonomic units of choice present relative to the abundance of each taxonomic entity. The index ranges from 1 for infinite diversity to 0 representing no diversity. In the phylum ratio analysis, the total detected kingdoms (Bacteria and Archaea) were subdivided according to the phylogenetic pyramid into subpopulations and then expressed as ratios. The beta diversity is a measure of similarity or dissimilarity between samples and groups of samples. Beta diversity was assessed between groups. The Bray–Curtis distance was used as a quantitative measure of abundance across different communities. Additionally, to quantitatively measure dissimilarities between groups, the Jaccard distance was calculated. The unweighted and weighted UniFrac indices were calculated from the phylogenetic tree as an indicator of dissimilarity between phylogenetic tree branches across glucose groups. Beta diversity was represented by 2D principal component analysis (PCoA) plots. The statistical significance in beta diversity measures between groups was calculated using PERMANOVA (PERmutational Multivariate ANalysis Of VAriance). Homogeneity of dispersion among groups was assessed using PERMDISP to ensure that observed differences reflected location rather than dispersion effects. Differential abundance (DA) analysis was performed to determine differentially abundant microbes at various taxonomic levels between the two groups. The method modeled each feature (e.g., an OTU, an organism, or species name) in a separate generalized linear model, where it was assumed that abundances follow a negative binomial distribution. The likelihood ratio test was used to determine significance across groups, and if significant, followed by the Wald test to determine significance between group pairs. The false discovery rate (FDR) was calculated to account for multiple testing [20,21]. Differential abundance results are presented in diagrams and quantitative tables.

For the demographic, clinical, and exercise variables, analyses were performed in SPSS (version 28), and *p* < 0.05 was considered significant. All analyses, including gut microbiome variables, were performed in CLC Genomics Workbench 20.0.3, QIIME2, and correction for multiple comparisons was performed with DRF, and a *q* > 0.05 was considered statistically significant.

## 3. Results

### 3.1. Exercise Distribution Across Glucose Groups

The distribution of participants across exercise arms was similar in the normoglycemic (Control 16, HIIT 15, MICT 13) and higher-risk (Control 10, HIIT 8, MICT 3) groups. A chi-square test indicated no significant difference between glucose categories (χ^2^(2) = 1.86, *p* = 0.39), confirming balanced exercise allocation. Accordingly, the observed microbial differences between glycemia groups are unlikely to be driven by exercise intervention effects.

### 3.2. Microbial Diversity and Composition at the Phylum Level

The results showed significant differences between the two (high and normal) glucose groups. The phylum-level alpha diversity, a measure of species richness and evenness within each individual sample, was estimated with the Shannon entropy and Simpson index. In general, a Shannon index at the phylum level of 0.004 to 2.94 is indicative of low alpha diversity. The score corresponds directly with sample heterogeneity. While the Simpson index scores range from 1 for infinite diversity to 0 representing no diversity. The mean species diversity score measured by Shannon and Simpson indices differed significantly between the normal and higher glucose groups (*p* = 0.0023 for the Simpson index and *p* = 0.005 for the Shannon index, as per Figure 3). However, larger inter-group dispersity of samples was observed in the normal glucose group, where outliers had a Simpson index ranging from as low as 0.26 to as high as 0.74. This might be due to the differences in glucose or demographic and lifestyle factors that are impossible to avoid in a study like this. The high glucose group had a tighter clustering (median ~0.52, smaller spread). The PERMDISP test, however, found that this difference in dispersion was *not statistically significant* (*F* = 2.33, *p* = 0.143) as per Appendix A. In the nonparametric Spearman correlation test, we confirmed significant correlations between glucose levels as continuous values across all participants and alpha diversity scores measured with Shannon and Simpson indices at the phylum level. Significant negative correlations were observed between fasting glucose and alpha diversity (Shannon: ρ = −0.341, *p* = 0.005; Simpson: ρ = −0.323, *p* = 0.009), indicating that higher glucose levels corresponded with lower microbial heterogeneity. These results confirm that higher glucose levels correspond with lower microbial heterogeneity. To further investigate group differences between the normal and higher glucose groups, beta diversity was measured and represented in 2D principal component analysis (PCoA) plots. The percentages in the brackets depicted in Figure 4 indicate how much of the total variation in the measured distance matrix is explained by each axis (PCo1 and PCo2). These two dimensions capture all the variations in the data. The first coordinate explained 81% of the variance, with the PCo2 explaining 19% of the variance, confirming that the glucose status was the dominant factor shaping microbiome differences.

We then analyzed the relative abundances of different bacterial phyla in the normal and high glucose groups (Figure 5). Phylum compositional distribution reveals the differences in Proteobacteria (3% in normal glucose versus 2% in higher glucose, which was statistically significant as per differential abundance analysis, with FDR corrected *p*-value of <0.05) (see Appendix A). Another significant change was reported in the relative abundance of phylum Verrucomicrobia. In the normal glucose group, Verrucomicrobia made up 5% of total microbes, whilst it was significantly lower in the high glucose group at 3% of total microbes (FRD *p*-value 0.04). The percentage of total Euryarchaetoa was also significantly lower in the high glucose compared to the low glucose group (FDR *p*-value of 0.03).

### 3.3. Differential Abundance Analysis at the Species Level

To examine in more detail the specific species, we conducted an analysis of the selective differential abundance of species previously reported to be associated with high glucose levels. We tested differential abundance across glucose groups using a feature-wise generalized linear model with a negative binomial mean–variance relationship. For each feature (OTU/species), an NB-GLM (negative binomial generalized linear model) was fitted with group as the main factor and library-size normalization via model-estimated size factors. Pairwise contrasts were evaluated with Wald tests, and the global (ANOVA-like) effect of group was assessed with a likelihood ratio test (LRT) comparing the full model to a reduced model without the group term. Multiple testing was controlled using the Benjamini–Hochberg false discovery rate (FDR) procedure (<0.05), following standard guidance on high-dimensional inference [22]. Differential abundance analysis with full OTU table (ANOVA-like comparison with FDR corrected *p*-values, also known as *q*-values of <0.05) amongst glucose groups revealed that 964 species differed between the normal and high glucose groups (Figure 6). Differential abundance analysis was performed using a negative binomial generalized linear model, which accounts for differences in sequencing depth and group sizes (n = 44 normal glucose; n = 21 high glucose). Dispersion estimates for included taxa were within acceptable ranges, indicating that variance was appropriately modeled despite the unequal group sizes.

Differential abundance for species of interest is listed in Table 1, showing species that are changed significantly between the glucose groups (glucose cut levels 4–5.4 mmol/L < or = normal and >5.5 mmol/L higher glucose. Normal n = 44 and higher risk n = 21).

To zoom into more details at the specific species level, we conducted selective differential abundance analysis of species previously reported to be associated with high glucose levels (results shown in Table 1).

We observed distinct patterns of microbial Beta diversity between the high- and normal glucose groups. Overall, the microbial diversity was significantly reduced in the high glucose group. At the highest taxonomic level (phylum), we found that Synergistes, Elusimicobia, Euryarchaeota, Verrucomicrobia, and Proteobacteria were all significantly lower in participants with high blood glucose (see Appendix A). Additionally, *P. copri* was significantly elevated in the high glucose group (10-fold increase, with FDR-corrected *p*-value < 0.001), suggesting that it may serve as an early inflammatory and diabetic marker. We also found that the *Fusobacterium* genus was significantly increased in the normal glucose group, with a 151-fold increase compared to the high glucose group (*p* < 0.005).

## 4. Discussion

This study relied on 16S rRNA gene sequencing, which characterizes microbial composition but does not provide functional or metabolic information. As such, the mechanistic links between microbial alterations and glucose metabolism remain speculative. Future studies combining metagenomic, metatranscriptomic, or metabolomic approaches are warranted to elucidate microbial functional capacities and their metabolic effects on host glucose regulation. The current findings should therefore be interpreted as descriptive and hypothesis-generating, pending validation in studies with greater analytical depth. In this Generation 100 sub-study, we found statistically significant distinct patterns in the gut microbiome between high and normal glucose groups. The alpha diversity was significantly lower in the high glucose group, confirming previous findings that the gut microbiome heterogeneity correlates with glucose levels [23,24]. Furthermore, the non-parametric Spearman correlation test revealed significant negative correlations between glucose levels and alpha diversity scores (Shannon and Simpson indices at the phylum level), confirming additional support for a link between higher glucose and lower microbial heterogeneity.

The gut epithelial barrier is maintained by a healthy, diverse gut microbiome composed primarily of four phyla: Bacteriodetes, Firmicutes, Actinobacteria, and Proteobacteria. A healthy gut microbiome consists of species assisting eubiosis, that is, supporting immune responses and maintaining a strong epithelial-mucosal barrier. We have compared microbial composition at the phylum level with ranges considered normal as per Ortiz-Alvarez et.al. 2020 [25] (Appendix A). We report a significantly lower ratio of Proteobacteria in the high glucose group compared to the normal glucose group. This contrasts with the findings of Ortiz-Alvarez et al., who reported substantially higher relative abundances (approximately 5–10%) in their study population [25]. However, Proteobacteria is a very large and diverse phylum that includes many important bacterial groups—some pathogenic (such as Escherichia coli, Salmonella, and Helicobacter pylori) and others beneficial or neutral. Therefore, it is vital to investigate the microbiome at lower taxonomic levels to distinguish which members of the phylum Proteobacteria are beneficial and which are harmful for glucose maintenance. Moreover, variations in findings across studies may suggest that factors other than blood glucose (e.g., diet, lifestyle, and geographic location) contribute to differences in the abundance of Proteobacteria. Gut bacteria are divided into three enterotypes according to their nutrient uptake. For example, enterotype II and III (Prevotella and Ruminococcus) are mucin degraders, whilst enterotype I (Bacteroides) utilizes energy from carbohydrates using glycolysis and pentose phosphate pathways [26]. In short, the abundance and role of Proteobacteria are shaped not only by blood glucose but also by diet, lifestyle, and geography, with different enterotypes specializing in distinct nutrient utilization strategies. Our total population cohort showed a larger abundance of Bacterioidetes (31–32%) in comparison with the population described by Ortiz-Alvarez et.al (20–25% of the total phyla). This phylum is composed of Gram-negative bacteria, many of which are involved in breaking down complex carbohydrates and other organic materials. A higher level of Bacteroidetes in the gut was also found to be associated with increased insulin resistance in humans in a recent clinical study [27,28]. Firmicutes include a wide range of bacteria, many of which are Gram-positive. It includes various species involved in fermentation processes and can play a role in energy absorption in the gut. A genus member of the phylum Firmicutes, *Christensenella*, was significantly increased 2.92 times in the normal glucose group compared with the high glucose group. Enrichment of genus *Christensenella* was reported to alleviate T2DM by promoting GLP-1 secretion, regulating hepatic glucose metabolism, inhibiting intestinal glucose absorption, enhancing intestinal barrier, reducing inflammation via LPS/TLR4/NF-κB pathway, and improving liver metabolism [29]. Specifically, a study by Wei-Shan Ang et.al [29] showed excellent properties of *C.minuta* as a potential probiotic therapy for metabolic diseases such as T2DM. Surprisingly, the normal glucose groups showed increased abundance of genus *Fusobacterium* compared to the high glucose group, whilst other studies found the presence of this bacterium in both oral and gut microbiome as correlated with T2DM and insulin resistance [30]. One possible explanation is that the ecological role of *Fusobacterium* may differ in the aging gut. Older adults often experience physiological changes, including reduced mucosal immunity, slower intestinal transit, and increased exposure to medications, which collectively alter microbial niches and interspecies interactions. In this context, *Fusobacterium* may not act primarily as a pro-inflammatory opportunist but rather reflect secondary ecological adaptation or oral–gut microbial translocation. Furthermore, differences in cohort characteristics, sequencing depth, and geographic diet patterns could contribute to these discrepancies. Future longitudinal and metagenomic studies will be necessary to clarify the functional relevance of *Fusobacterium* in glucose regulation within older populations. The relatively small and uneven sample size (21 hyperglycemic vs. 44 normoglycemic participants) represents a key limitation of this study and may reduce statistical power for detecting subtle microbial differences. Consequently, the associations observed (such as between *Fusobacterium* and glucose levels) should be interpreted cautiously. Nonetheless, these findings provide preliminary insights that warrant confirmation in larger, independent cohorts. The finding that *Fusobacterium* was more abundant in the normal glucose group, despite its reported enrichment in T2DM cohorts, may be explained by several factors. First, differences in population characteristics—particularly age, diet, and medication use—can strongly influence microbial composition and may lead to distinct outcomes across studies. Second, *Fusobacterium* is a heterogeneous genus comprising species and strains with diverse ecological roles, ranging from commensal to opportunistic. Third, the effect of *Fusobacterium* likely depends on its microbial context; in a balanced gut community, its activity may be moderated by protective taxa, whereas in metabolically dysregulated states, it may contribute to inflammation. Moreover, the association observed in T2DM may reflect correlation rather than causation, as *Fusobacterium* could simply proliferate under conditions characteristic of disease rather than actively drive pathology. Finally, methodological differences—including sequencing region, analytical pipeline, and cohort size—may also contribute to divergent findings. These considerations underscore the importance of interpreting *Fusobacterium*-related results within the demographic and ecological context of each study population.

Although participants were drawn from an exercise intervention trial, exercise exposure was balanced across glycemia groups (χ^2^(2) = 1.86, *p* = 0.39). Furthermore, microbial patterns associated with exercise did not correspond with those identified between glucose categories. Therefore, the gut microbiome differences reported here are unlikely to be confounded by exercise status. Importantly, having well-defined exercise exposure represents a strength of this study, as most published microbiome–glycemia studies lack such information. Collectively, these analyses indicate that while residual confounding by exercise cannot be fully excluded, it is unlikely to represent a major limitation in this dataset. Notably, having well-characterized exercise exposure represents a strength of this study compared with many observational microbiome–glycemia studies where physical activity is typically unreported and uncontrolled. Differential abundance analysis shows unique microbial patterns in high and normal glucose groups. The microbial composition at the phylum level showed 2.5% lower Verrucomicrobia levels in the high glucose group, whilst *Akkermansia muciniphila*, a member of the phylum Verrucomicrobia, was 29-fold higher in the normal glucose group. Verrucomicrobia is a relatively lesser-known phylum of bacteria, but it includes some important microorganisms, such as *Akkermansia muciniphila*, which is present in the human gut and is associated with beneficial effects on metabolism and gut health. Verrucomicrobia are generally Gram-negative and can be found in diverse environments, including soil, water, and the human intestine. *Akkermansia* spp. play a role in glucose metabolism by improving insulin sensitivity and glucose tolerance. The bacterium has been shown to decrease fasting blood glucose levels and improve insulin resistance [31]. Overall, several other studies associate the presence of *Akkermansia* spp. with improved metabolism, decreased inflammation, and improved intestinal barrier function [32]. A study in diet-induced mice confirmed the beneficial effect of pasteurized *Akkermansia muciniphila*, where supplementation of this bacterial species increased energy expenditure and enhanced physical activity [33]. Oral administration of the *A. muciniphila* to high-fat-diet-mice increased glucose tolerance and reduced adipose tissue inflammation [34]. This genus has gained special attention due to its involvement in mucin degradation and ability to regulate genes of other bacterial species, creating a symbiotic homeostasis environment in the gut mucosa.

Our study shows a significant increase in *Akkermansia* spp. suggesting beneficial probiotic potential in reducing T2DM or prediabetes risk. Dietary strategies to increase *A. muciniphila* abundance in gut microflora exist along with commercially available probiotics [35]. Other bacteria from the phylum Bacteroidota named *Prevotella copri* were significantly elevated (10-fold increase) in the high glucose group, suggesting that it may serve as an early diabetic marker. These findings are consistent with previous research identifying *P. copri* as a pro-inflammatory pathogen [36,37].

Previous studies show that higher levels of enterobacteriaceae are associated with poor glycemic control, metabolic syndromes including obesity, insulin resistance, and impaired lipid profile [38,39]. The most recent metadata analysis by Gurung et al. from 2020 summarizes 42 human studies in which T2D was investigated in terms of the gut microbiome [38]. According to their results, genera that were negatively associated with T2D were: Bifidobacterium, Bacteroides, Faecalibacterium, Akkermansia, and Roseburia, and higher abundance/level of these species is associated with lower T2DM risk. In our study, we did not find significant differences in the relative abundance of Bifidobacterium, Faecalibacterium, or Bacterioidetes between the high and normal blood glucose groups. This discrepancy may reflect differences in cohort characteristics and study design. Our participants were older adults without diagnosed diabetes, in whom age-related declines in *Bifidobacterium* and *Faecalibacterium* may already be present across the cohort, thereby limiting between-group variation. Moreover, 16S rRNA V3–V4 sequencing offers limited resolution for distinguishing closely related species within these genera.

This study was a sub-study of the Generation 100 Study, and hence, uneven sample sizes per glucose groups were present (n = 44 vs. n = 21). This imbalance mainly reduces power in the smaller group and can inflate type-I error in PERMANOVA if dispersion differs—hence the beta dispersity was checked within the group. In future research, the use of Bayesian models could help to incorporate prior knowledge to stabilize inference in small groups. The major limitation of this study, however, lies in the fact that only those willing and eligible (after 5 years of participation + MRI eligibility + microbiome consent) were included, which may not be representative of the main cohort or the population at large. Older adults with health issues, dementia, or MRI contraindications were excluded. Additionally, as explained earlier, this study is a secondary analysis in an exercise RCT. Thus, blood glucose differences may be confounded by exercise exposure, sex, cohabitation, or other baseline factors related to the main study. However, where applicable, we used multivariate regression models adjusting for age, sex, exercise group allocation, and BMI.

Overall, a sub-study like this is a great way to utilize all the data from expensive, large studies (RCTs) to ensure all information is extracted and analyzed. Various multivariate prediction models could be applied in the future to decipher various confounders and their effect on the total microbiome. The next step for future research on the significance of microbiome in better glucose metabolism is to design large global cohort studies, which would cover a larger population, offering better representation of the geographic and ethnic differences. Additionally, a large prospective microbiome intervention study would be a valuable next step, using strains identified in recent research as a probiotic mixture to evaluate their modulatory effects on blood glucose in patients with prediabetes or T2DM. Larger, prospective cohorts in which microbiome, glucose, and lifestyle factors are measured longitudinally would provide an excellent opportunity to advance our understanding of the impact of the microbiome on glucose management. In particular, the use of causal inference frameworks will be essential to distinguish correlation from causation, which in this field of research is especially challenging to disentangle. The present study focused on compositional profiling of the gut microbiota using 16S rRNA gene sequencing to identify taxa associated with glycemic variation in older adults. As such, functional or causal mechanisms could not be directly inferred. Nevertheless, several genera identified here (such as *Akkermansia*, *Prevotella*, and *Fusobacterium*) have been previously implicated in glucose metabolism, mucosal integrity, and inflammatory regulation, suggesting possible metabolic relevance of these compositional patterns. Future analyses will integrate metagenomic and metabolomic data from the same biobanked samples to explore microbial functional pathways and metabolite profiles that may underlie these associations. Such work will help determine whether the observed microbial shifts contribute to, or result from, alterations in glucose homeostasis.

## 5. Conclusions

This study uniquely focuses on an older adult cohort, a demographic often overlooked in microbiome–metabolism research. By examining glycemic variation within this population, we tried to map the early microbial patterns that may reflect age-related metabolic adaptation or dysregulation. These insights could inform future interventions aimed at preserving metabolic health through microbiome modulation in aging. The present 16S rRNA analysis identifies compositional differences in the gut microbiota associated with glucose status, providing preliminary evidence of microbial shifts relevant to metabolic regulation. While 16S sequencing does not capture functional or mechanistic details, several of the taxa identified here—such as *Akkermansia* and *Fusobacterium*—have been functionally characterized in previous metagenomic and metabolomic studies. Accordingly, our interpretations are based on these established findings and should be viewed as associative rather than causal. Future analyses incorporating metabolomic data from this cohort will help to elucidate the microbial functional pathways underlying these associations. In accord with prior research, we observed reduced microbial diversity and distinct compositional shifts in individuals with elevated fasting glucose, reinforcing the concept that dysbiosis is a hallmark of early type 2 diabetes development. Importantly, our findings identify specific taxa—such as the striking elevation of *P. copri* and the unexpected enrichment of Fusobacterium in normoglycemic individuals—that may represent early microbial signatures of glucose dysregulation or protection. By situating these observations within the prediabetic stage, this study adds to the growing evidence that microbiome alterations precede overt diabetes, thereby underscoring their potential as biomarkers for early detection and as targets for preventive interventions. Moreover, this study contributes to the growing body of evidence on the role of the gut microbiome in type 2 diabetes. Collectively, the findings highlight the potential for identifying novel bacterial species that may serve as indicators of dysbiosis. Understanding these microbial shifts not only deepens our knowledge of disease mechanisms but also opens avenues for interventions aimed at restoring microbial balance, thereby improving glucose sensitivity and metabolic outcomes.

## Figures and Tables

**Figure 1 microorganisms-13-02582-f001:**
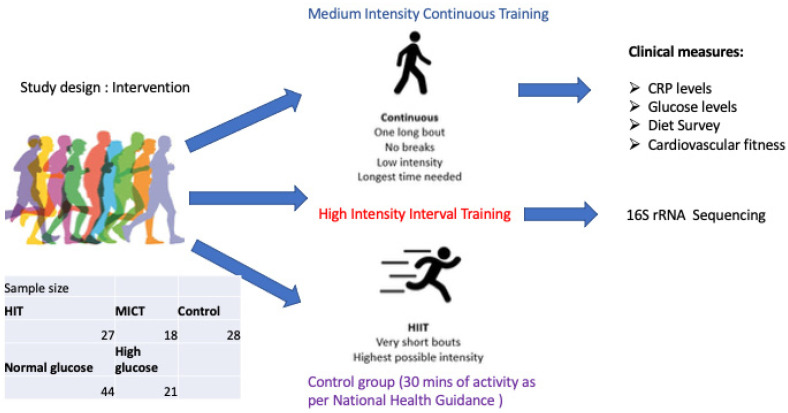
Generation 100 sub-study design.

**Figure 2 microorganisms-13-02582-f002:**
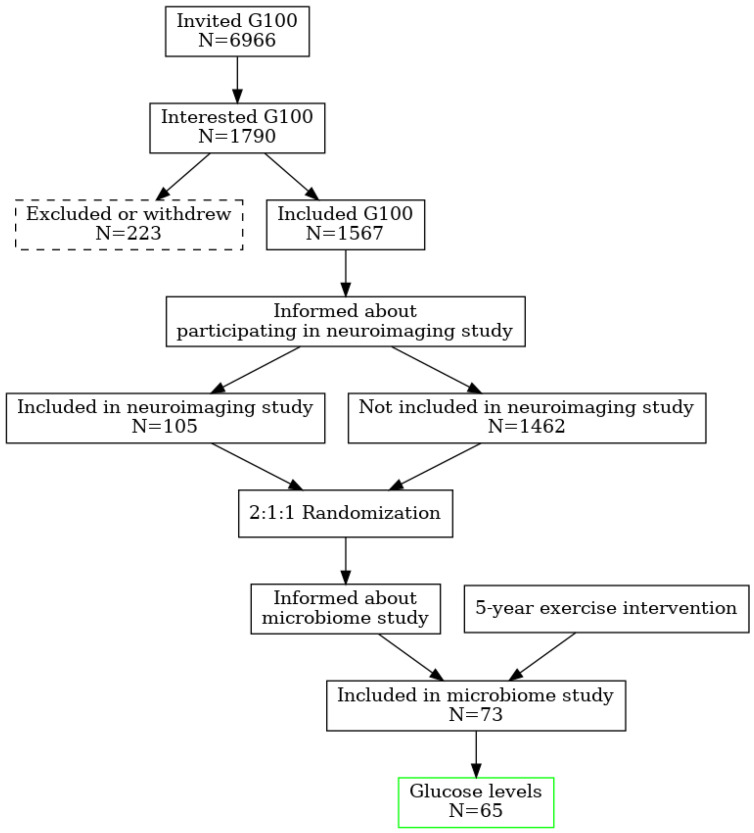
Study design and randomization flowchart.

**Figure 3 microorganisms-13-02582-f003:**
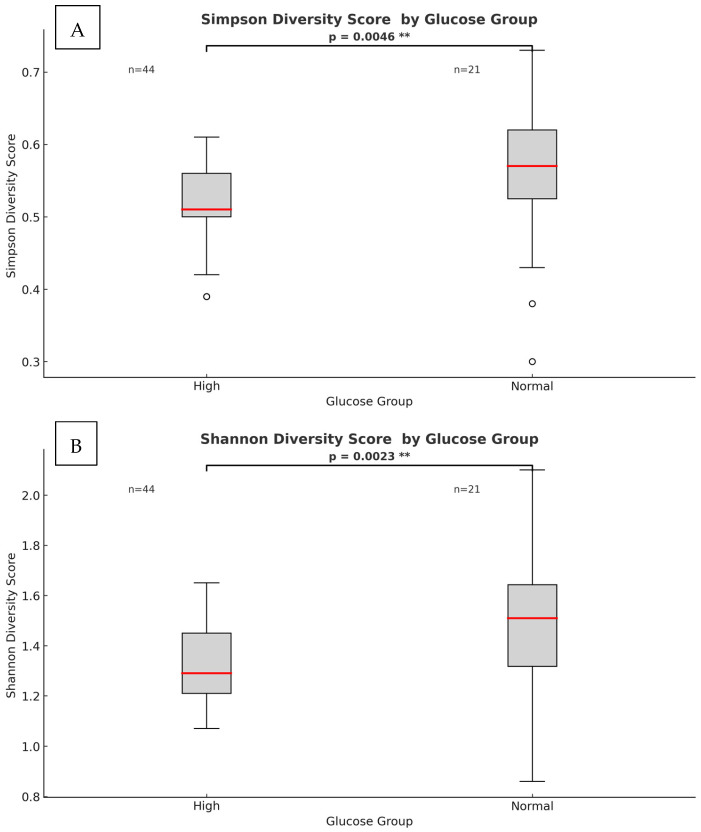
**Phylum-level heterogeneity between high and normal glucose groups.** (**A**) Simpson index α-diversity measure (*p* = 0.0046). (**B**) Shannon index α-diversity measure (*p* = 0.0023). The red line within each box represents the median value (50th percentile) of the diversity scores, while the box edges indicate the interquartile range. Points beyond the whiskers represent outliers. (**) indicates significant *p*-value.

**Figure 4 microorganisms-13-02582-f004:**
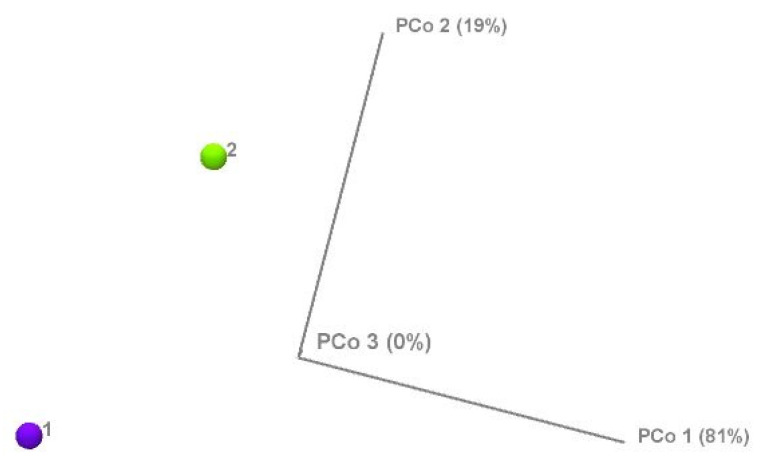
Beta diversity of the normal glucose (1) and high glucose (2) groups measured by principal component analysis.

**Figure 5 microorganisms-13-02582-f005:**
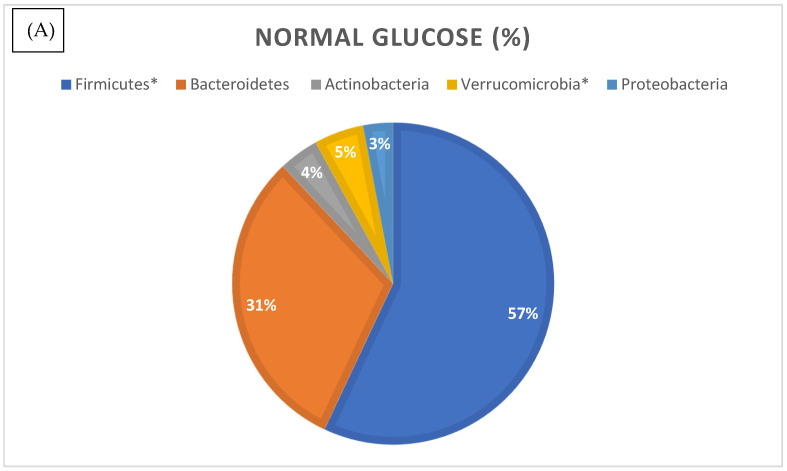
Relative distribution of bacterial phyla identified by 16S rRNA gene sequencing in the normal (**A**) and higher glucose (**B**) groups. Phyla marked with an asterisk (*) were significantly different in abundance between groups (*p* < 0.05, differential abundance analysis).

**Figure 6 microorganisms-13-02582-f006:**
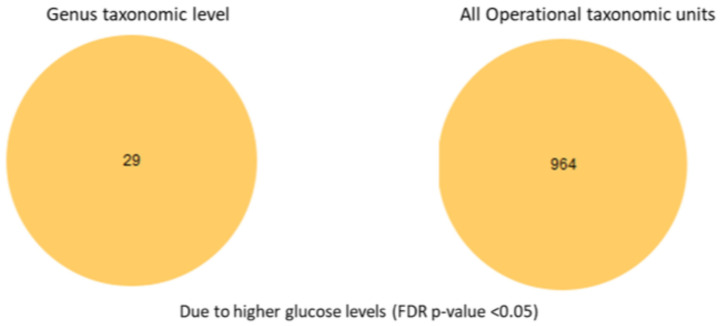
Differential abundance analysis at the genus taxonomic level and all taxonomic unit levels.

**Table 1 microorganisms-13-02582-t001:** Differential abundance analysis with full OTU table (ANOVA-like comparison with FDR corrected *p*-values/*q*-values < 0.05) between normal versus high glucose groups. (*) were significantly different in abundance between groups (*p* < 0.05, differential abundance analysis).

Changed Due to Glucose Levels in G100 Sub-Study (Comparison of Low Glucose vs. High Glucose)			
Species Associated Negatively with T2D, a ‘’Healthy’’ Microbiome Species			
	Fold Change	*p*-Value	FDR *p*-Value/*q*-Value
*Bifidobacterium*	N/A		
*Faecalibacterium*	N/A		
*Akkermansia (genus)*	3.17	3.61 × 10^−3^	0.01 *
*s__muciniphila*, 192963	29.51	1.80 × 10^−6^	0.0000148 *
*Roseburia (genus)*	N/A		
*Christensenella (genus)*	2.92	0.02	0.05 *
f__Christensenellaceae, 549577	3.58	0.02	0.05 *
Species positively associated with T2D, ‘’unhealthy’’ microbiome			
*Prevotella*	2.05	0.28	0.48 *
s__copri, 909561	−1.06 × 10^3^	<0.001	<0.001 *
*Ruminococcus*	N/A		
*Fusobacterium*	141.47	5.56 × 10^−8^	0.000000495 *
*Blautia* spp.	N/A		
Other previously not reported species that were changed due glucose levels in G100 Study			
Fusobacterium	151.93	5.52 × 10^−8^	0.000000446 *
*Citrobacter*	184.62	1.36 × 10^−11^	0.000000000242 *
*Klebsiella*	59.9	1.27 × 10^−7^	0.000000942 *
*Catenibacterium*	21.59	6.49 × 10^−5^	0.000304 *
*Enterococcus*	6.54	0.01	0.04 *
*Odoribacter*	N/A		
*Escherichia*	14.06	2.84 × 10^−4^	0.0012 *
*Sutterella*	−7.11	0.03	0.06 *
*Citrobacter*	184.62	1.36 × 10^−11^	0.000000000242 *
*Peptococcus*	602.29	1.69 × 10^−12^	0.0000000000375 *
*Methanobrevibacter*	4.45	9.88 × 10^−3^	0.03 *
*Serratia*	−270.48	8.40 × 10^−3^	0.02 *

## Data Availability

The data could be obtained via a request to the Generation 100 Study database. We are also happy to provide raw data upon request to the corresponding author.

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
