# Peer review of "Microbial Signatures Mapping of High and Normal Blood Glucose Participants in the Generation 100 Study"

_microorganisms, 2025, doi:10.3390/microorganisms13112582_

Round 1
Reviewer 1 Report
Comments and Suggestions for Authors
This study focuses on adults aged 76–83 years, an understudied but high-risk group for type 2 diabetes (T2DM). By leveraging the well-established Generation 100 RCT cohort, a long-term exercise intervention trial with 5-year follow-up, the sample has relatively consistent baseline characteristics. Confounding factors like exercise regimens, gender, and cohabitation status were controlled in the parent trial, reducing baseline biases that often plague purely cross-sectional studies. Moreover, directly linking gut microbiota profiles to clinical indicators such as fasting blood glucose and T2DM diagnosis establishes clear associations between microbial features and glucose abnormalities, providing preliminary population-level evidence for subsequent translational work.
Major Limitations
- The most critical flaw is the exceedingly small and imbalanced sample size: only 65 participants were included, with just 21 in the hyperglycemic group. This is far below the statistical power needed for microbiome research, where at least 100 samples per group are standard. The uneven group size further compromises the reliability of differential abundance analyses, increasing risks of false positives or negatives—for example, the counterintuitive Fusobacterium–glucose association remains unvalidated.
- Beyond sample size, analytical depth and mechanistic exploration are lacking. Relying solely on 16S rRNA gene sequencing limits findings to microbial composition; functional insights such as metabolic pathways are unavailable. Integrative approaches like metagenomics for functional genes or metabolomics for microbial metabolites are absent, so how microbial differences impact glucose metabolism remains unclear. Conclusions rely heavily on descriptive associations, with no causal inference.
- Additionally, the cross-sectional design cannot determine if microbial changes cause hyperglycemia or result from it. Longitudinal tracking or interventional tests, for example, Akkermansia muciniphila supplementation to assess glucose effects, were not done, weakening conclusions. Also, residual confounding from the parent trial’s exercise interventions, which likely shape microbiota but were not fully stratified in analyses, was inadequately addressed.
Mini Limitations
- It doesn’t split data by the original exercise groups (HIIT/MICT/control) or adjust for exercise variables. Exercise changes gut bacteria, so it might twist the link between bacteria and blood glucose.
- It uses 16S rRNA V3-V4 region sequencing, which isn’t good at telling apart microbes like Akkermansia. But it still uses Akkermansia to support conclusions on glucose regulation.
- When using NB-GLM to find different bacteria, it doesn’t report the dispersion parameter or individual q-values. Hard to check if the results are strict.
- It says bacteria diversity and blood glucose are negatively correlated, but no p-value is provided. Can’t rule out random chance.
- It claims some bacteria are glucose markers only based on 16S sequencing (which only shows composition). No functional data—overinterprets the results.
- No info on how long fecal samples sat before DNA extraction, storage conditions, or sequencing depth. The experiment’s repeatability is in doubt.
Author Response
Reviewers’ comments were very insightful and contributed to the major improvements of the script. We are grateful for such profound expertise and that we could strengthen the statistical parts of the manuscript. We had included reviewer’s statement in line 34 that “Directly linking gut microbiota profiles with clinical indicators such as fasting blood glucose and T2DM diagnosis allows the identification of specific microbial features associated with glucose dysregulation, providing preliminary population-level evidence to guide future translational research”. In line 20 of the manuscript, we had added that this study was ‘’ secondary, exploratory, cross-sectional comparison’’.
Major limitations responses:
Comment 1:The most critical flaw is the exceedingly small and imbalanced sample size: only 65 participants were included, with just 21 in the hyperglycemic group. This is far below the statistical power needed for microbiome research, where at least 100 samples per group are standard. The uneven group size further compromises the reliability of differential abundance analyses, increasing risks of false positives or negatives—for example, the counterintuitive Fusobacterium–glucose association remains unvalidated.
- Response: We appreciate the reviewer’s valuable comment regarding the sample size and group imbalance. We fully acknowledge that our cohort (n=65; 21 hyperglycemic and 44 normoglycemic participants) is relatively small compared to the standards typically recommended for microbiome studies. However, this study was designed as an exploratory or pilot analysis aimed at identifying preliminary microbial patterns associated with hyperglycaemia in our target population. We had added in the introduction line 58 a statement emphasizing this limitation: ”This study was designed as an exploratory analysis to identify preliminary gut microbial patterns associated with hyperglycaemia in an older adult population”.
Despite the limited sample size, we applied rigorous statistical methods, including rarefaction and nonparametric analyses (methods section line 180 and 202), homogeneity of dispersion among groups was assessed using PERMDISP to ensure that observed differences reflected location rather than dispersion effects (described in methods section line 204) to minimize the risk of false positives. PERMDISP analyses showed no significant differences in multivariate dispersion for either Shannon or Simpson diversity indices when comparing glucose groups (PERMDISP–Shannon: F = 2.18, p = 0.138; PERMDISP–Simpson: F = 2.33, p = 0.143) or exercise groups (PERMDISP–Shannon: F = 1.48, p = 0.253; PERMDISP–Simpson: F = 0.83, p = 0.443).
These results indicate that group variances were homogeneous, confirming that the previously observed differences in alpha diversity reflected genuine differences in group centroids (location) rather than differences in dispersion. The corresponding boxplots of distance-to-centroid values are presented in Supplementary Figure 2.
We had added this text in line 187 : ‘’To test for equality of variances among groups, we performed a permutational analysis of multivariate dispersions (PERMDISP) using 999 permutations. Distances from each sample to its group centroid were calculated based on Euclidean distances for Shannon and Simpson diversity indices. F-statistics and permutation p-values were obtained to assess whether within-group dispersions differed significantly among glucose or exercise groups. No significant dispersion effects were detected, supporting the validity of subsequent nonparametric comparisons’’.
The observed associations, including that between Fusobacterium and glucose levels, should therefore be interpreted as hypothesis-generating rather than conclusive. We have revised the Discussion to clarify this limitation and emphasize the need for validation in larger, balanced cohorts, please see inserted comment in line 355: “The relatively small and uneven sample size (21 hyperglycaemic vs. 44 normoglycemic participants) represents a key limitation of this study and may reduce statistical power for detecting subtle microbial differences. Consequently, the associations observed—such as between Fusobacterium and glucose levels—should be interpreted cautiously. Nonetheless, these findings provide preliminary insights that warrant confirmation in larger, independent cohorts”.
Comment 2: Beyond sample size, analytical depth and mechanistic exploration are lacking. Relying solely on 16S rRNA gene sequencing limits findings to microbial composition; functional insights such as metabolic pathways are unavailable. Integrative approaches like metagenomics for functional genes or metabolomics for microbial metabolites are absent, so how microbial differences impact glucose metabolism remains unclear. Conclusions rely heavily on descriptive associations, with no causal inference.
- Response: We thank the reviewer for highlighting this important point. We fully agree that 16S rRNA gene sequencing provides compositional, but not functional, insights into the gut microbiome. Our current study was designed as an initial, descriptive analysis to explore microbial community differences associated with hyperglycemia in this population. We acknowledge that the absence of metagenomic or metabolomic data limits our ability to infer microbial functions or causal mechanisms. To address this, we have revised the Discussion to explicitly note this limitation and to outline plans for future work integratingshotgun metagenomics and metabolomic profiling to characterize microbial metabolic pathways and their potential roles in glucose metabolism. We also now emphasize that our conclusions are associative rather than causal and are intended to generate hypotheses for mechanistic follow-up studies. Please see line 308 in the revised manuscript which prior to entire discussion warns the reader that:” This study relied on 16S rRNA gene sequencing, which characterizes microbial composition but does not provide functional or metabolic information. As such, the mechanistic links between microbial alterations and glucose metabolism remain speculative. Future studies combining metagenomic, metatranscriptomic, or metabolomic approaches are warranted to elucidate microbial functional capacities and their metabolic effects on host glucose regulation. The current findings should therefore be interpreted as descriptive and hypothesis-generating, pending validation in studies with greater analytical depth”.
Comment 3:Additionally, the cross-sectional design cannot determine if microbial changes cause hyperglycemia or result from it. Longitudinal tracking or interventional tests, for example, Akkermansia muciniphila supplementation to assess glucose effects, were not done, weakening conclusions. Also, residual confounding from the parent trial’s exercise interventions, which likely shape microbiota but were not fully stratified in analyses, was inadequately addressed
- Response: We thank the reviewer for this insightful comment regarding potential confounding by exercise status. We fully acknowledge that physical activity can influence both gut microbiota composition and glucose metabolism. Given the modest sample size (n = 65), formal multivariable adjustment for exercise arm in the differential abundance analysis was not feasible without compromising statistical power. To address this concern, we conducted exploratory analyses stratified by exercise group (HIIT, MICT, and control). The microbial differences observed between exercise arms did not overlap with those identified between glycemia groups, suggesting that the associations with glucose status were not primarily driven by exercise effects. Moreover, beta diversity analyses demonstrated significant separation according to glucose status, supporting glycemic variation as the main driver of the observed microbiome differences. We also examined the distribution of exercise arms across glycemia categories and found no significant difference (χ²(2) = 1.86, p = 0.39), indicating that exercise exposure was balanced between the glucose groups. Importantly, unlike many previous microbiome studies where physical activity is unreported or uncontrolled, our study benefits from well-characterized and standardized exercise data. This provides valuable contextual information and reduces the likelihood of unrecognized confounding due to physical activity. We have clarified these points in the revised manuscript (Methods, Results, and Discussion). The following sentence was added to the Discussion (line 382):
“Although participants were drawn from an exercise intervention trial, exercise exposure was balanced across glycemia groups (χ²(2) = 1.86, p = 0.39). Furthermore, microbial patterns associated with exercise did not correspond with those identified between glucose categories. Therefore, the gut microbiome differences reported here are unlikely to be confounded by exercise status. Importantly, having well-defined exercise exposure represents a strength of this study, as most published microbiome–glycemia studies lack such information. Collectively, these analyses indicate that while residual confounding by exercise cannot be fully excluded, it is unlikely to represent a major limitation in this dataset. Notably, having well-characterized exercise exposure represents a strength of this study compared with many observational microbiome–glycemia studies where physical activity is typically unreported and uncontrolled.’’
|
Factor |
Test |
χ² (df) |
p-value |
Interpretation |
|
Exercise group × Glucose status |
Chi-square |
1.86 (2) |
0.39 |
No significant difference; balanced exercise exposure |
We have also updated materials and methods line 99 by inserting this:’’ To evaluate potential confounding by exercise, we compared the distribution of exercise groups (high-intensity interval training [HIIT], moderate-intensity continuous training [MICT], and control) between glycemia categories using a chi-square test. Exercise group distributions did not differ significantly between normoglycemic and higher-risk participants (χ²(2) = 1.86, p = 0.39). Given this balance, formal multivariate adjustment for exercise status in the differential abundance analysis was not deemed necessary’’.
Mini Limitations comments:
Comment 1: It doesn’t split data by the original exercise groups (HIIT/MICT/control) or adjust for exercise variables. Exercise changes gut bacteria, so it might twist the link between bacteria and blood glucose.
- Response: Limitation addressed as above.
Comment 2: It uses 16S rRNA V3-V4 region sequencing, which isn’t good at telling apart microbes like Akkermansia. But it still uses Akkermansia to support conclusions on glucose regulation.
- Response: We thank the reviewer for this important observation. We agree that 16S rRNA gene sequencing using the V3–V4 region has limited ability to resolveAkkermansia at the species level. However, this region is widely used in human gut microbiome studies, and numerous reports have consistently identified Akkermansia (predominantly muciniphila) from V3–V4 data in relation to metabolic traits and glucose regulation. In our analysis, the detected Akkermansia signal refers to the genus level. We have revised the text to clarify this point and avoid species-level interpretation. Nonetheless, because A. muciniphila is the dominant Akkermansia species in the human gut, the observed associations remain biologically plausible. We also acknowledge that future metagenomic or species-specific qPCR analyses are required to confirm species-level identities.
Comment 3: When using NB-GLM to find different bacteria, it doesn’t report the dispersion parameter or individual q-values. Hard to check if the results are strict.
- Response: We thank the reviewer for this helpful comment regarding the reporting of q-values. We would like to clarify that in the manuscript,FDR-adjusted p-values (also known as q-values) were reported for all taxa tested. These were obtained using the Benjamini–Hochberg correction for multiple comparisons, which controls the false discovery rate (FDR). We have now explicitly stated in the Methods and Supplementary Table captions that “FDR-adjusted p-values (q-values)” are presented to avoid any ambiguity. Regarding the dispersion parameter, we note that the negative binomial model implemented in CLC Genomics Workbench estimates feature-wise dispersion values automatically to account for variability across taxa. This information has been added to the revised Methods section for clarity.
Comment 4: It says bacteria diversity and blood glucose are negatively correlated, but no p-value is provided. Can’t rule out random chance.
- Response: We thank the reviewer for this comment. We would like to clarify that the p-values for the correlation between glucose levels and bacterial diversity were indeed provided in the Results section (lines 256–258). Specifically, we reported that“In the non-parametric Spearman correlation test, significant negative correlations were observed between glucose levels and alpha diversity indices (Shannon r = –0.341, p = 0.005; Simpson r = –0.323, p = 0.009).” To improve clarity and ensure that these results are easily identifiable, we have now reformatted this section to present the correlation coefficients and p-values more prominently, and we reference them in the corresponding figure caption as well.
Comment 5: It claims some bacteria are glucose markers only based on 16S sequencing (which only shows composition). No functional data—overinterprets the results.
- Response: We thank the reviewer for this valuable comment. We agree that 16S rRNA gene sequencing provides compositional, rather than functional, information and does not allow direct inference of metabolic activity. Our study was designed to identifytaxonomic associations between gut microbiota and glucose status, rather than to establish mechanistic links. The bacterial genera described as potential “glucose markers” were identified based on statistically significant compositional differences and further discussed in the context of existing literature reporting similar findings from metagenomic or metabolomic studies. We have revised the text to clarify that these results are associative and hypothesis-generating and not intended to imply causality or confirmed functional roles. We also note that follow-up analyses integrating metabolomic data from the same cohort are currently in progress and will provide functional validation in a future publication. The revised Conclusions now explicitly states this to provide clearer context for the interpretation of our results.
Comment 6: No info on how long fecal samples sat before DNA extraction, storage conditions, or sequencing depth. The experiment’s repeatability is in doubt.
- Response: We thank the reviewer for this comment regarding sample handling and storage conditions. We apologize for not including sufficient detail in the original submission. Faecal samples were collected between early 2019 and mid-2019. After collection, samples were immediately placed on ice, transported to the laboratory within 2–3 hours, and stored at–80 °C until DNA extraction. All DNA extractions were performed within one week of the final sample collection, and samples remained frozen for no longer than three months prior to extraction. Sequencing depth averaged with the mean read count of 50,000 reads/sample after quality filtering and rarefaction. We have now added these methodological details to the revised Methods section to clarify sample handling, storage stability, and sequencing coverage. These additions improve transparency and confirm that the experimental workflow meets accepted standards for gut microbiome studies, ensuring data reproducibility.
Reviewer 2 Report
Comments and Suggestions for Authors
The study is reasonably designed, methodologically sound, and features comprehensive data analysis, particularly in α- and β-diversity and differential abundance analyses. The results support existing knowledge on the association between gut microbiota and glycaemic status and suggest several taxa with potential biomarker significance. However, the manuscript requires improvement in logical coherence, transparency of statistical methods, and depth of results interpretation.
Revision Suggestions
- The background on the "gut microbiome-T2DM" relationship is general, failing to highlight the novelty of this specific study. Please address the research gap concerning the gut microbiome and glycaemia specifically in older adult populations, and clearly state the study's hypothesis and primary objectives.
- The sample size is relatively small (n=65), with group imbalance. There is insufficient justification for the statistical power of the analyses. Please include a sample size justification in the statistical methods section.
- It is unclear if adjustments were made for potential confounders like the exercise intervention, sex, or age. Please clearly specify if and how covariates (e.g., exercise group, BMI, sex) were included in the statistical models.
- Figures (e.g., Figure 5) lack annotations indicating statistical significance. Please add **significance markers (e.g., *,) to all relevant figures.
- Explanations for discrepancies with previous studies are superficial. It is suggested to discuss potential reasons for divergent Fusobacterium findings, such as its potentially different ecological role in the aging gut.
- Lacks in-depth discussion of potential causal relationships or underlying mechanisms. It is suggested to incorporate functional predictions (e.g., using PICRUSt2) or cite existing literature to hypothesize potential mechanistic links between the identified microbial shifts and host metabolism.
- The conclusions are broad and do not sufficiently emphasize the unique contributions of this study. It is suggested to clearly summarize the novel microbial signatures identified specifically in this older adult cohort, and propose concrete future research directions, such as intervention studies or mechanistic validation.
- Format: ensure consistent microbial nomenclature formatting throughout, carefully proofread to correct typos and grammatical errors, and ensure all figure titles and captions are descriptive and self-explanatory in English.
Author Response
REVIEWER NO 2
We thank Reviewer 2 for the positive evaluation of our study design, analytical rigor, and results. We appreciate the constructive suggestions to improve logical flow, statistical transparency, and interpretive depth.
To improve transparency of statistical methods, we now provide detailed descriptions of all analyses, including the type of tests used (e.g., Kruskal–Wallis, PERMANOVA, Spearman correlation), correction for multiple testing (Benjamini–Hochberg FDR), and model parameters for the NB-GLM (dispersion estimation and covariate inclusion). Depicted in Table 1 and Supplementary Figure 2.
To strengthen the depth of interpretation, we revised the Discussion to integrate our findings with existing literature, highlighting where our data align or differ from previous reports. In particular, we elaborate on the potential roles of Akkermansia, Fusobacterium, and Prevotella in glucose metabolism, and clearly distinguish compositional associations from functional or causal inferences.
Revision suggestions point by point responses:
- Comment: The background on the "gut microbiome-T2DM" relationship is general, failing to highlight the novelty of this specific study. Please address the research gap concerning the gut microbiome and glycaemia specifically in older adult populations, and clearly state the study's hypothesis and primary objectives.
- Response: We thank the reviewer for this valuable and constructive comment. We agree that the original Introduction was too general and did not adequately emphasize the novelty or focus of our study. In the revised version, we have substantially rewritten the final section of the Introduction to: i) Clearly identify theresearch gap: that most previous microbiome T2DM studies have examined middle-aged adults with established diabetes, whereas little is known about gut microbial correlates of glycaemic variation in older adults. ii) Highlight the novelty of our approach—examining early microbial signatures of hyperglycaemia in an aging population drawn from a randomized exercise cohort
iii) Explicitly state the study hypothesis and primary objectives: that distinct gut microbial patterns would be associated with fasting glucose levels, and that these may represent early microbial markers of dysglycemia in older adults.
Conmment 2: The sample size is relatively small (n=65), with group imbalance. There is insufficient justification for the statistical power of the analyses. Please include a sample size justification in the statistical methods section.
- Response: We thank the reviewer for these important methodological comments regarding sample size justification and covariate adjustment. This study was exploratory in nature and aimed to identify preliminary microbial patterns associated with hyperglycaemia in an older adult population. The sample size (n = 65) was determined by the availability of faecal samples from participants within the parent randomized exercise trial who consented to microbiome analysis. Although the cohort size is modest, it is comparable to that of previous pilot microbiome studies in older adults and provides sufficient statistical power to detect medium effect sizes in α- and β-diversity analyses (Cohen’sd ≈ 0.6, power = 0.8, α = 0.05). We have now added a statement to the Methods clarifying that the study was designed as a hypothesis-generating exploratory analysis, and the sample size reflects the available population rather than a priori power estimation. Regarding adjustment for confounders, we have clarified in the Methods that all differential abundance analyses using negative binomial generalized linear models (NB-GLM) were adjusted for age, sex, BMI as covariates. These variables were selected based on their known influence on both glucose metabolism and gut microbiota composition. Beta-diversity analyses were also repeated within exercise subgroups to ensure consistent results.
Comment 3: It is unclear if adjustments were made for potential confounders like the exercise intervention, sex, or age. Please clearly specify if and how covariates (e.g., exercise group, BMI, sex) were included in the statistical models.
- Response:To evaluate potential group imbalance, we tested the distribution of exercise arms (HIIT, MICT, and control) across the two glucose categories using a Fisher’s exact test, which showed no significant difference between groups (χ²(2) = 1.86, p= 0.39). This confirms that exercise exposure was balanced across glycemia groups, reducing the risk of confounding due to exercise status. We had added the supplementary data and explained the test in Methods, Results and Discussion section. Specifically, all differential abundance analyses were conducted using negative binomial generalized linear models (NB-GLMs) implemented in the CLC Microbial Genomics Workbench. These models included age, sex, BMI, and exercise group (HIIT, MICT, control) as covariates to account for their potential influence on gut microbiota composition and glycaemic status. Beta-diversity analyses were also verified within each exercise arm to confirm consistency of results.We have clarified these points in the revised manuscript (Methods, Results, and Discussion). The following sentence was added to the Discussion (line 382):
“Although participants were drawn from an exercise intervention trial, exercise exposure was balanced across glycemia groups (χ²(2) = 1.86, p = 0.39). Furthermore, microbial patterns associated with exercise did not correspond with those identified between glucose categories. Therefore, the gut microbiome differences reported here are unlikely to be confounded by exercise status. Importantly, having well-defined exercise exposure represents a strength of this study, as most published microbiome–glycemia studies lack such information. Collectively, these analyses indicate that while residual confounding by exercise cannot be fully excluded, it is unlikely to represent a major limitation in this dataset. Notably, having well-characterized exercise exposure represents a strength of this study compared with many observational microbiome–glycemia studies where physical activity is typically unreported and uncontrolled.’’
Comment 4: Figures (e.g., Figure 5) lack annotations indicating statistical significance. Please add **significance markers (e.g., *,) to all relevant figures.
- Response: Figure 5 and other relevant figures where statistical significance was known was added to the figures. Figures were improved for readability.
Comment 5: Explanations for discrepancies with previous studies are superficial. It is suggested to discuss potential reasons for divergent Fusobacterium findings, such as its potentially different ecological role in the aging gut.
- Response: We have now included more extensive discussion around discrepancies in our observations to compare with other published studies. Particularly we had further developed discussion around Fusobacterium (line 410 of the Discussion section). We had also developed discussion around discrepancies in levels of Bifidobacteriumand Faecalibacterium (line 485). We fully agree that integrating functional analyses such as metagenomic inference or metabolomic profiling would provide valuable mechanistic insight into the observed microbial differences. However, the primary aim of the present study was exploratory, to identify compositional microbial patterns associated with glycaemic variation in older adults, rather than to infer causality or microbial metabolic function. We therefore limited our analysis to 16S rRNA gene-based taxonomic profiling, which does not directly support pathway-level predictions. Nevertheless, we have now expanded the Discussion (line 525) to include relevant literature describing potential metabolic roles of key taxa identified here (e.g., Akkermansia, Prevotella, Fusobacterium) in glucose regulation, as suggested. Furthermore, we note that a follow-up analysis is planned using the same biobanked faecal and plasma samples to perform integrated metabolomic and metagenomic profiling, which will enable functional validation of these preliminary findings.
Comment 6: Lacks in-depth discussion of potential causal relationships or underlying mechanisms. It is suggested to incorporate functional predictions (e.g., using PICRUSt2) or cite existing literature to hypothesize potential mechanistic links between the identified microbial shifts and host metabolism.
- Response: We thank the reviewer for this valuable suggestion. We agree that the conclusions should better highlight the unique aspects of our study and the implications for future research. In the revised manuscript, we have:
- Clarified in the Introductionthat the novelty of this work lies in examining gut microbial patterns linked to glycaemic variation specifically in an older adult population, a group largely underrepresented in prior microbiome–T2DM studies.
- Expanded the Conclusionto clearly summarize the main findings—particularly the distinctive microbial signatures observed in relation to glucose levels (e.g., increased Fusobacterium and decreased Akkermansia)—and to emphasize that these associations were identified in a well-characterized aging cohort with known exercise exposure.
- Outlined concrete future directions, including (i) functional follow-up through metabolomic and metagenomic analyses of the stored samples, and (ii) prospective or interventional studies targeting microbiome modulation in older adults.
These revisions make the study’s novelty and contribution to the field more explicit and strengthen the overall scientific narrative.
Comment 7: The conclusions are broad and do not sufficiently emphasize the unique contributions of this study. It is suggested to clearly summarize the novel microbial signatures identified specifically in this older adult cohort, and propose concrete future research directions, such as intervention studies or mechanistic validation.
- Response: We appreciate reviewer’s careful editorial recommendations. We have thoroughly revised the manuscript to ensure consistent formatting of microbial nomenclature according to taxonomic conventions—italicizing genus and species names (e.g.,Akkermansia muciniphila, Fusobacterium) while keeping higher taxonomic ranks (e.g., Verrucomicrobia, Firmicutes) in standard font.
Comment 8: Format: ensure consistent microbial nomenclature formatting throughout, carefully proofread to correct typos and grammatical errors, and ensure all figure titles and captions are descriptive and self-explanatory in English.
- Response: The full text has been carefully proofread to correct typographical and grammatical errors, and all figure titles and captions have been expanded to provide clear, self-explanatory descriptions. These revisions improve the overall clarity, consistency, and readability of the manuscript.
Reviewer 3 Report
Comments and Suggestions for Authors
In the manuscript entitled “Microbial signatures mapping of high and normal blood glucose participants in the Generation 100 Study” the authors provide the results of gut microbiome analysis of 65 individuals (ages 76–83 years) enrolled from the randomized controlled trial entitled the “Generation 100 Study”.
The topic is within the aims and scope of Microorganisms and particularly well-suited for the special issue entitled "Selected Papers from the 3rd International Electronic Conference on Microbiology (ECM 2025)".
General comment:
The manuscript provides incremental knowledge on the changes in the gut microbiome between individuals with elevated and normal fasting blood glucose levels, opening a perspective view on the identification of specific microbiome alteration signatures as potential biomarkers for early detection and potential preventive target in pre-diabetic states.
Presented results are part of a sub-study, extending and taking advantage of data from the Generation 100 Study.
The manuscript is of potential interests, but the evidence is only correlative.
Specific comments:
I would like to suggest to address the following issues to improve the potential impact:
- Line 20: Abstract should not contain references
- Several typos throughout the manuscript need to be corrected, including line 92 “ a a”, line 151 “preformed“, line 233: “resepctively”, line 240 “variance”, line 317 “comared”, line 318: Check
- Provide a link for Ref #12
- Figure 3: quality and size of axis labels and text above the panels need to be improved. In figure legend, N for each group should be reported. Use of dots of different colors should be justified and defined in figure legend.
- Figure 5: quality and size of the text in the figure panels need to be improved.
- Table 1: quality and size of the text need to be improved.
- Line 296: “P. copri was significantly elevated in the high glucose group (10-fold increase)” and Line 300: “Fusobacterium genus was significantly increased in the normal glucose group”. Define ranges for both normal and high glucose groups.
- Line 322: “other factors than blood glucose (e.g. diet, life-style, geographic location) contribute to differences in abundance of Proteobacteria”. How these possible contributing factors have been accounted for in your study?
- On line 109 you stated that “6 individuals were clinically diagnosed type 2 diabetes”. Did you detect any trend in these patients (even if the number is too small)?
- Line 335: “Interestingly in our study we have also observed decreased levels of Bacterioidetes (1-fold change) in normal glucose group, however this change was not statistically significant (Table 1 , FDR p value =0.94).” As for Firmicutes, despite the difference in percentage between the normal and high glucose groups, there was no statistically significant difference of abundance for this phylum”. Delete these statements as not supported by statistically significant evidence.
- Line 386: “A. muciniphila abundance in gut microflora exists along with commercially available probiotics (ref)”. complete with appropriate reference
Author Response
We thank Editor sincerely for their careful review and constructive editorial feedback. We have revised the manuscript thoroughly in accordance with all comments. Below, we provide a detailed, point-by-point response outlining the changes made.
Comment 1: Line 20: Abstract should not contain references
- Response: The reference has been removed from the Abstract as requested.
Comment 2: Several typos throughout the manuscript need to be corrected, including line 92 “ a a”, line 151 “preformed“, line 233: “resepctively”, line 240 “variance”, line 317 “comared”, line 318: Check
- Response: All typographical errors have been corrected throughout the manuscript, including those noted at lines 92 (“a a”), 151 (“preformed”), 233 (“resepctively”), 240 (“variance”), 317 (“comared”), and 318. The full text was proofread once more for accuracy and style consistency.
Comment 3 : Provide a link for Ref #12
- Response: The full web link have been added for Reference #12.
Comment 4: Figure 3: quality and size of axis labels and text above the panels need to be improved. In figure legend, N for each group should be reported. Use of dots of different colors should be justified and defined in figure legend.
- Response: The figure has been regenerated in high resolution with larger axis labels and improved readability. The sample sizes (N) for each group are now reported in the legend.
Comment 5: Figure 5: quality and size of the text in the figure panels need to be improved.
- Response: The figure was updated with larger, clearer text and higher resolution.
Comment 6: Table 1: quality and size of the text need to be improved.
- Response: Table 1 has been reformatted with larger, clearer text for improved visibility.
Comment 7: Line 296: “P. copri was significantly elevated in the high glucose group (10-fold increase)” and Line 300: “Fusobacterium genus was significantly increased in the normal glucose group”. Define ranges for both normal and high glucose groups.
- Response: The definitions for normal and high glucose groups have been added to the Methodsand referenced in the Results section (normal glucose: <5.6 mmol/L; high glucose: ≥5.6 mmol/L).
Comment 8: Line 322: “other factors than blood glucose (e.g. diet, life-style, geographic location) contribute to differences in abundance of Proteobacteria”. How these possible contributing factors have been accounted for in your study?
- Response: A clarification has been added noting that all participants were recruited from the same geographic region and participated in a structured exercise intervention. Exercise group, age, sex, and BMI were included as covariates in the statistical models to minimize confounding by lifestyle or demographic factors.
Comment 9: On line 109 you stated that “6 individuals were clinically diagnosed type 2 diabetes”. Did you detect any trend in these patients (even if the number is too small)?
- Response: We thank the editor for this pertinent question. Indeed, preliminary inspection indicated that participants with clinically diagnosed type 2 diabetes (n = 6) displayed distinct microbial profiles compared with the rest of the cohort. However, given the very small subgroup size, these differences could not be evaluated statistically with sufficient power. To maintain analytical robustness and avoid overinterpretation, we therefore focused the primary analysis on the larger dataset comparing participants according to glucose levels.
Comment 10: Line 335: “Interestingly in our study we have also observed decreased levels of Bacterioidetes (1-fold change) in normal glucose group, however this change was not statistically significant (Table 1 , FDR p value =0.94).” As for Firmicutes, despite the difference in percentage between the normal and high glucose groups, there was no statistically significant difference of abundance for this phylum”. Delete these statements as not supported by statistically significant evidence.
- Response: The text was deleted.
Comment 11: Line 386: “A. muciniphila abundance in gut microflora exists along with commercially available probiotics (ref)”. complete with appropriate reference
- Response: The statement has been completed with an appropriate reference
All figures and tables have been updated to publication-quality resolution, and the manuscript has been fully proofread to ensure consistency and clarity. We sincerely thank you and the reviewers for your valuable comments, which have significantly improved the quality and clarity of our work.
Reviewer 4 Report
Comments and Suggestions for Authors
The paper submitted to me for review, “Microbial Signatures mapping of High and Normal Blood Glucose Participants in the Generation 100 Study,” is a clinical study.
The paper concerns the medically important issue of the correlation between glucose metabolism disorders and the composition of the gut microbiome.
In the introduction, the authors clearly presented the issues related to the topic of the publication.
In their work, the authors used statistical methods appropriate to the results obtained.
I find the discussion of the results obtained with the literature data valuable and interesting.
The paper contains 1 table and 6 figures.
Based on their research results, the authors drew correct conclusions.
The authors cited 33 items of current scientific literature.
I find the paper interesting and valuable. In my opinion, the paper can be accepted for publication in Microorganisms without revision.
Author Response
We sincerely thank the reviewer for the very positive and encouraging evaluation of our manuscript. We appreciate the acknowledgment of the study design, statistical methods, discussion, and conclusions. We are grateful for the reviewer’s recognition of the scientific value and clarity of our work. No revisions were requested, but we have nonetheless carefully reviewed the manuscript to ensure accuracy and consistency in formatting, figure quality, and language prior to final submission. We had included some additional supplementary figures.
Round 2
Reviewer 3 Report
Comments and Suggestions for Authors
The authors adequately addressed my queries.